# Cu(I) Complexes of Multidentate *N,C,N-* and *P,C,P-*Carbodiphosphorane Ligands and Their Photoluminescence

**DOI:** 10.3390/molecules25173990

**Published:** 2020-09-01

**Authors:** Marius Klein, Nemrud Demirel, Alexander Schinabeck, Hartmut Yersin, Jörg Sundermeyer

**Affiliations:** 1Department of Chemistry and Science, Materials Sciences Center, Philipps University of Marburg, 35043 Marburg, Germany; kleinma8@staff.uni-marburg.de (M.K.); Demireln@students.uni-marburg.de (N.D.); 2Institute for Physical Chemistry, University of Regensburg, 93040 Regensburg, Germany; alexander.schinabeck@chemie.uni-regensburg.de (A.S.); Hartmut.Yersin@chemie.uni-regensburg.de (H.Y.)

**Keywords:** carbodiphosphorane, phosphorus ylides, pincer ligands, coordination chemistry, Cu(I) complex, photoluminescence

## Abstract

A series of dinuclear copper(I) *N,C,N-* and *P,C,P-*carbodiphosphorane (CDP) complexes using multidentate ligands CDP(Py)_2_ (**1**) and (CDP(CH_2_PPh_2_)_2_ (**13**) have been isolated and characterized. Detailed structural information was gained by single-crystal XRD analyses of nine representative examples. The common structural motive is the central double ylidic carbon atom with its characteristic two lone pairs involved in the binding of two geminal L-Cu(I) fragments at Cu–Cu distances in the range 2.55–2.67 Å. In order to enhance conformational rigidity within the characteristic Cu–C–Cu triangle, two types of chelating side arms were symmetrically attached to each phosphorus atom: two 2-pyridyl functions in ligand CDP(Py)_2_ (**1**) and its dinuclear copper complexes **2–9** and **11**, as well as two diphenylphosphinomethylene functions in ligand CDP(CH_2_PPh_2_)_2_ (**13**) and its di- and mononuclear complexes **14**–**18**. Neutral complexes were typically obtained via the reaction of **1** with Cu(I) species CuCl, CuI, and CuSPh or via the salt elimination reaction of [(CuCl)_2_(CDP(Py)_2_] (**2**) with sodium carbazolate. Cationic Cu(I) complexes were prepared upon treating **1** with two equivalents of [Cu(NCMe)_4_]PF_6_, followed by the addition of either two equivalents of an aryl phosphine (PPh_3,_ P(C_6_H_4_OMe)_3_) or one equivalent of bisphosphine ligands bis[(2-diphenylphosphino)phenyl] ether (DPEPhos), 4,5-bis(diphenylphosphino)-9,9-dimethylxanthene (XantPhos), or 1,1′-bis(diphenyl-phosphino) ferrocene (dppf). For the first time, carbodiphosphorane CDP(CH_2_PPh_2_)_2_ (**13**) could be isolated upon treating its precursor [CH(dppm)_2_]Cl (**12**) with NaNH_2_ in liquid NH_3_. A protonated and a deprotonated derivative of ligand **13** were prepared, and their coordination was compared to neutral CDP ligand **13**. NMR analysis and DFT calculations reveal that the most stable tautomer of **13** does not show a CDP (or carbone) structure in its uncoordinated base form. For most of the prepared complexes, photoluminescence upon irradiation with UV light at room temperature was observed. Quantum yields (*Φ*_PL_) were determined to be 36% for dicationic [(CuPPh_3_)_2_(CDP(Py)_2_)](PF_6_)_2_ (**4**) and 60% for neutral [(CuSPh)_2_(CDP(CH_2_PPh_2_)_2_] (**16**).

## 1. Introduction

In 1961, hexaphenyl-carbodiphosphorane, the first carbodiphosphorane (CDP), was synthesized by Ramirez et al. [1]. Despite this early discovery, the interest in such double ylide carbon (or carbone) compounds is still evolving. One reason for attracting interest is the bonding description of carbodiphosphoranes. Next to a classical ylide valence bond description, the bonding in carbodiphosphoranes can be decribed as a formal carbon(0) atom stabilized by two dative phosphine ligands with C–P retro dative bonding components, which is a model discussed earlier but quantified by a theoretical approach of Frenking and co-workers [2,3,4,5,6]. The central carbon atom is best described in its excited singlet (^1^D) state [7]. It acts as an acceptor and is stabilized by the σ donating phosphine ligands. The two characteristic occupied lone pairs (HOMO and HOMO+1) centered at this carbon atom (therefore named “carbone”) are either capable of binding two metals via two σ bonds in a close to tetrahedral configuration P_2_CM_2_ or one metal in a trigonal–planar P_2_CM configuration via a σ- and a π dative bond of very strong π,σ-donor character [8]. For this reason, the coordination chemistry of carbodiphosphoranes has experienced a renaissance [9,10,11]. A topic of current interest is introducing secondary ligand functions into the CDP frame: Cyclometalation with noble metals rhodium and platinum gave rise to the characterization of *C*,*C*,*C*-pincer ligand complexes with two cyclometalated phenyl rings [12,13,14,15,16,17], and an ortho-directed double lithiation of hexaphenyl-carbodiphosphorane leads to lithium complexes that are capable of transfering the *C,C,C*-pincer ligand synthon [CDP]^2−^ to any other element of the periodic table [17]. *P*,*C*,*P*-chelate complexes of a phosphine functionalized CDP ligand CDP(CH_2_PPh_2_)_2_ (**13**), formally a carbone C(dppm)_2_ (dppm = bis-diphenylphosphinomethane), were characterized, but the free ligand **13** was not isolated so far [18,19,20,21,22,23,24]. Only recently, complexes of 2-pyridyl functionalized *N*,*C*,*N*-carbodiphosphorane CDP(Py)_2_ (**1**) have been reported [25,26]. The isolation of the free ligand base **1 [25]** enabled the synthesis of Cu(I) CPD complexes, which are discussed in this work. Cu(I) complexes [27,28,29,30,31,32,33,34,35,36,37,38,39,40,41,42,43] can be used as cost-efficient luminescent materials, which potentially can replace highly phosphorescent Ir [44,45,46,47,48,49,50] or Pt [47,51,52,53,54,55,56,57,58] complexes in OLED technology. For example, OLED devices with internal quantum efficiencies of up to 100% could be realized based on the thermally activated delayed fluorescence *(TADF*) *singlet-harvesting* mechanism [28,29,30,31,41]. According to this mechanism, both the singlet and triplet excitons formed in an OLED emission layer can be harvested, and emission occurs via the S_1_ state. 

Very frequently, Cu(I) complexes exhibit low-lying metal-to-ligand charge transfer (MLCT) transitions that are related to small energy separations *ΔE*(S_1_–T_1_) between the lowest singlet S_1_ and the lowest triplet T_1_ state due to small HOMO–LUMO overlap. As a consequence, efficient up-intersystem crossing (T_1_→S_1_), also designated as reverse intersystem crossing RISC, can occur at near ambient temperature [28,41,45,59,60], thus resulting in thermally activated delayed fluorescence (TADF). This is also related to a small transition dipole moment, and thus, a small radiative rate *k^r^*(S_1_→S_0_) [31,32]. The described MLCT formally corresponds to the oxidation of Cu(I) to Cu(II) and leads to photo-induced structural rearrangements in the excited state(s) being connected to large Franck–Condon factors [61], and as a consequence, to competing non-radiative relaxations. Therefore, the design of rigid structures with small reorganization energy between the ground state and excited states is essential. 

While the first luminescent behavior of an Au(I) N-heterocyclic carbene (NHC) complex was already described in 1999 [62], it took another 10 years until the first photoluminescent Cu(I) NHC complexes were characterized [63] followed by further studies more recently [64,65,66,67,68,69,70]. In contrast to the π-acidic NHCs ligands, the π-donating CDP ligands have not yet been considered in luminescent materials. Herein, we report such luminescent Cu(I) CDP complexes, their synthesis, X-ray structure data, and photoluminescence properties. We demonstrate, that high emission quantum yields can be obtained with selected materials of this class.

## 2. Results

### 2.1. Synthesis and Characterization of N,C,N-CDP Complexes

The *N*,*C*,*N*-carbodiphosphorane pincer ligand CDP(Py)_2_ (**1**) was synthesized as reported previously [25] and used as a ligand in order to synthesize neutral and cationic dinuclear copper (I) complexes. Complexes **2**, **3**, and **9** were conveniently prepared by stirring ligand **1** with two equivalents of the respective copper(I) salts CuX in THF at room temperature for 18 h. Moderate yields of 86% and 63% for **2** and **3**, as well as 27% for **9** were achieved in form of orange powders. Dicationic complexes **4**–**8** were prepared in an in situ two-step protocol by the reaction of CDP(Py)_2_ (**1**) with tetrakis(acetonitrile)copper(I) hexafluorophosphate (2 eq.) in THF, followed by the addition of either two equivalents of monodentate triaryl phosphine or one equivalent of a bisphosphine ligand: triphenylphosphine, tris(*o*-methoxyphenyl)phosphine, bis[(2-diphenylphosphino)phenyl] ether (DPEPhos), 4,5-bis(diphenylphosphino)-9,9-dimethylxanthene (XantPhos), and 1,1’-bis(diphenyl-phosphino) ferrocene (dppf) were chosen as ligands. The dicationic Cu(I) complexes were isolated and crystallized in yields of 47–90% (Scheme 1). Additionally, a neutral Cu(I) CDP complex was obtained via the deprotonation of carbazole (**10**) in THF using sodium *tert*-butoxide and the addition of [(CuCl)_2_(CDP(Py)_2_)] (**2**) to this solution. [(CuCarb)_2_(CDP(Py)_2_)] (**11**) was obtained as light orange powder in a yield of 56%. Complexes **2**–**9** and **11** have been characterized via ^31^P{^1^H} NMR, ^1^H-NMR, ^13^C{^1^H} NMR, and by elemental analyses. Due to the typically poor volatility of ionic and zwitterionic Cu(I) complexes **2**–**9** and **11**, no mass spectra with molecular ions were obtained under EI, FD, and ESI ionization techniques.

Single crystals suitable for X-ray diffraction analysis were obtained upon layering THF or DCM solutions of the complexes with *n*-pentane. Crystal structures for **2**, **4**, **6**, **7**, **9**, and **11** are shown in Figure 1, selected bond distances and angles are shown in Table 1. Further details of the XRD analyses of **3**, **5**, and **8** are described in the Appendix A. The molecular structures of **2**–**9** reveal that the central carbon atom within the CDP ligand is capable of coordinating two copper atoms in a geminal fashion. Each copper atom is additionally coordinated by one 2-pyridyl unit of ligand **1**. If the Cu–Cu interaction is disregarded, the two copper atoms per molecule are coordinated in a planar fashion, which is more T-shaped than trigonal planar. Each copper atom is interacting with one of the two carbone lone pairs of the central carbon atom C1, each by one nitrogen atom of a 2-pyridyl chelate ring and by the variable neutral ligand L or anionic ligand X. The strongest ligand interactions (C and X/phosphine) define a Cu(I) archetypical close to the linear axis. The geminal nature of both copper(I) centers leads Cu–Cu distances in the range of 2.55–2.67 Å (Table 1). These distances are smaller than twice the size of the covalent radius of Cu (1.32 Å) [71] or twice the size of the van der Waals radius of Cu (1.4 Å) [72]. Twice the size of the Cu(I) covalent radius (1.27 Å) [73] is close to the observed Cu–Cu distance. Similar trends are observed in dinuclear Cu(I) CDP complexes without any constraints of additional chelating CDP functions [74]. The Cu–Cu interaction leads to a formally coordinatively saturated pseudo tetrahedral coordination around each copper atom. This dinuclear entity is intramolecularly stabilized by a neutral 4-electron donor carbone ligand bridging the two Cu atoms. This rather rigid ligand template is characterized by characteristic torsion angles X–Cu–Cu–X in the range 41.9° (**2**)–76.0° (**3**) for anionic ligands X (X = Cl, I, S(C_6_F_6_) or L–Cu–Cu–L in the range 62.4° (**8**)–82.9° (**4**) for phosphine and the bridging bisphosphine ligands. The rather rigid frame of this *N,C,N*-ligand backbone seems to be privileged to stabilize this 8-electron-5-center inner Cu_2_CN_2_ core.

Representative parent complex **2** crystalizes in a triclinic crystal system with a crystallographic point group of *P*-1 and with four units and two unique molecules in the unit cell. One of the two independent molecules is slightly disordered, and both have very similar geometric parameters. The angles (°) around copper are almost identical for the two Cu atoms, but crystallographically, they are not strictly identical: C–Cu–Cl 162.11(8)°, C–Cu–N 89.45(9)°, and N–Cu–Cl 106.95(6)°. Each copper atom deviates only marginally from the plane defined by C, N, and X = Cl to which copper(I) is bound. Cu–Cu distances, which indicate weak Cu–Cu interactions, e.g., 2.5525(5) Å for **2**. The C–Cu–Cu angles of 2 and related species are typically sharp, e.g., 49.98(7)° in case of **2**. A comparable coordination scenario can be found for the other complexes **3**–**9**. Only small differences for the C–P distances as well for the Cu–C–Cu and the P–C–C angles are observed within the series **2**–**9**.

Complex **11** crystalizes in a monoclinic crystal system with a space group of *P*2_1_/*n* and four units in its unit cell. In contrast to the described XRD molecular structures of **2**–**9**, the neutral complex **11** shows only one pyridine copper interaction, while the remaining pyridyl unit stays in a dangling nonbonding situation. The carbazolyl anions display a perpendicular orientation with respect to each other. Both steric and electronic factors are probably responsible for the dangling pyridyl unit in **11**. As expected, the Cu–N_carb_ distance 1.911(3) Å for copper with the higher coordination number due to additional pyridine interaction is longer than Cu–N_carb_ 1.861(2) Å for the other one. According to NMR spectroscopy, there is a dynamic exchange process of bonded and dangling pyridine ligands in solution. 

### 2.2. Synthesis and Characterization of P,C,P–CDP Complexes

Peringer et al. developed *P*,*C*,*P*–CDP pincer complexes of a formal carbone ligand C(dppm)_2_, which was not isolated and characterized, but trapped in the form of its complexes [18,19,20,21,22,23,24]. The synthetic strategy involved complex redox reactions. It is limited to the characterization of Ni(II), Pd(II), Pt(II), or Au(III) complexes so far. Our synthetic approach was to isolate the free CDP base. Thus, [CH(dppm)_2_]Cl (**12**) [18,19] was treated with an excess of sodium amide (6.5 eq.) in liquid ammonia at −78 °C. Since the basicity of sodium amide leads to the deprotonation of only one proton, CDP(CH_2_PPh_2_)_2_ (**13**) could be isolated in 98% yield as an intense yellow powder. No further deprotonation products and no adduct formation with lithium salts were observed as in the case of using organolithium bases. The isolation of **13** was the precondition to access the coordination chemistry of Cu(I) with this *P*,*C*,*P*–CDP ligand base. Dinuclear copper complexes **14**–**16** were synthesized and characterized via NMR spectroscopy and mass spectrometry (Scheme 2). 

Upon treating **13** with tetrakis(acetonitrile)copper(I) hexafluorophosphate in DCM, a cationic complex [CuCl(H-CDP(CH_2_PPh_2_)_2_]PF_6_ (**17**) was obtained. The enhanced basicity of alkyl-substituted CDP **13** compared to pyridyl-substituted CDP **1** leads to a protonation of a Lewis acid-activated acetonitrile ligand. Therefore, monoprotonated **13** is acting as a ligand in mononuclear copper complex **17** with hexafluorophosphate as a counter ion. While searching for adequate bases for the deprotonation of **12**, we observed the ability of *n*-BuLi (2 eq.) to further deprotonate CDP **13**, generating an anionic CDP ligand **20** (Scheme 3) as lithium salt. Trapping this anion with one equivalent of tetrakis(acetonitrile)copper(I) hexafluorophosphate and one equivalent of triphenylphosphine leads to neutral copper(I) complex **18** as a light yellow powder in 73% yield. **18** was characterized via ^31^P{^1^H} NMR, ^1^H-NMR, and elemental and XRD analysis. 

After the deprotonation of symmetric protonated CDP form **12,**
^31^P{^1^H} NMR spectra of the product (or products) become temperature and solvent-dependent. We presumed that this observation could be an indication of the presence of more than one tautomer, at least two with definitely chemically non-equivalent ^31^P nuclei of monodeprotonated base **13** (see Appendix A). As there were no literature data available on this particular carbodiphosphorane **13**, even though it was used as a ligand in several publications, we decided to investigate the tautomeric forms of **13** via computational methods (Scheme 3). Geometry optimizations were performed at the PBE-D3(BJ)/def2-TZVPP level of theory, which were followed by single-point calculations and a natural bond orbital (NBO) analysis at the PBE0-D3(BJ)/def2-TZVPP level of theory. Interestingly, the results reveal that the free ligand base **13** cannot be acknowledged as a carbodiphosphorane, but rather as tautomer **13a**. Due to the high first proton affinity (PA) and drastically lower second PA of the alkyl-substituted central CDP carbon atom and due to the enhanced CH acidity of the methylene group placed in between a phosphanyl and a phosphionio functionality, the ground state of **13** is not represented by tautomer **13c** or **13b** but by asymmetric tautomer **13a**. This equilibrium explains the highly complex ^31^P{^1^H} NMR spectra obtained from solutions of pure **13**. Symmetric tautomer **13b** is 4.1 kcal/mol more stable than **13c**, but asymmetric **13a** is 7.7 kcal/mol more stable than **13b**. Therefore, **13b** seems to be observable at very low concentration in a dynamic equilibrium ratio next to **13a** but not symmetric carbodiphosphorane form **13c**. 

Our results from solution and gas phase investigation and very clear results from XRD solid-state investigations of ligand **13** complexes indicate that the equilibrium of tautomers displayed in Scheme 3 is shifted toward **13c**, if the free base **13** is trapped by coordination with two Cu(I) ions. The further deprotonation of **13a** leads to symmetric carbanion **20** as the most stable tautomer: **20a** with equally CH-functionalized C1, C2, and C3 is 12.7 kcal/mole more stable than asymmetric tautomer **20b** retaining a carbodiphosphorane structure. A hypothetical 1λ^5^,3λ^3^ diphosphete derivate **20c** is just 1.1 kcal/mole less stable than **20b** in the gas phase. The charge distribution of the tautomers can be monitored via NBO analysis. While the atomic partial charge *q*(C) of C1 of **13a** is −1.38 e, which corresponds to *q*(C) of the protonated hexaphenyl-carbodiphosphorane (−1.33 e) [6], the one of **13c** reveals as −1.45 e and therefore is in the same order of magnitude as for the hexaphenyl-carbodiphosphorane (−1.43 e) [6]. For **20a**, the *q*(C) values of C1, C2, and C3 are –1.39 e, 1.37 e, and 1.37 e, while the *q*(C) values of P1, P2, P3 and P4 are 1.68 e, 1.68 e, 0.83 e and 0.83 e. For more information regarding the atomic partial charges and for a detailed deprotonation of **12**, see Appendix A, as well as Appendix A in the Appendix A. 

Single crystals suitable for X-ray diffraction analysis were obtained upon layering a THF or a DCM solution of the complexes **14**–**18** with *n*-pentane. The XRD molecular structures are depicted in Figure 2, while selected bond distances and angles are shown in Table 2 and Table 3. For dinuclear complexes **14**–**16**, a very similar trend is observed, as discussed in Chapter 2.1. The central CDP carbon atom acts as 4-electron donor involving two geminal copper atoms into a Cu–C–Cu triangle. Each copper atom is further coordinated to one chelating phosphine group. While **14** and **16** crystalize in a triclinic crystal system with space group *P*-1 and two units in the unit cell, **15** crystalizes in a monoclinic crystal system with space group *C*2/c and four units in the unit cell. In contrast to dinuclear Cu(I) complexes of pyridyl-CDP **1**, complexes **14**–**16** of phosphanyl-CDP **13** reveal significantly longer Cu–Cu distances (Å). 2.8681(5) (**14**), 2.8816(12) (**15**), and 2.989(2) (**16**) compared to 2.5525(5) (**2**) and 2.671(2) (**11**). This is in accord with the higher steric demand of the phosphine and an increased freedom of motion in CDP ligand **13** compared to the more rigid and compact CDP **1** (also compare the XYZ.file of the SI). In contrast to **2**–**9**, disregarding the Cu–Cu interaction, a less pronounced T-shape but more trigonal planar coordination sphere of the copper(I) ions is observed for **14**–**16**. This is probably due to the fact that phosphines, carbones, and the anions X are more similar in their donor strength and Cu(I) affinity compared to weaker pyridine ligands in the first series of compounds. For **14**, the angles (°) around copper are 128.57(7) (C–Cu–Cl), 99.71(7) (C–Cu–P) and 129.61(3) (P–Cu–Cl) and therefore closer to the ideal 120° of a trigonal coordination sphere compared to **2**. This rather rigid ligand template is characterized by characteristic torsion angles X–Cu–Cu–X in the range 119.9° (**15**)–140.2 (**16**) and are therefore larger compared to the complexes of **2**. The less rigid frame of this *P,C,P* ligand backbone stabilizes an 8-electron-5-center inner Cu_2_CP_2_ core.

Selected bond distances and angles of **14**–**16** can be found in Table 2, which demonstrates an increase of the Cu–C–Cu angle of about 10° in addition to the increased Cu–Cu distances relating to the increasing freedom of motion of **13** compared to **1**. The P–C–P angles of the CDP complexes **14**–**16** are comparable to the ones of ligand **1**. 

Selected bond distances and angles of **17** and **18** are displayed in Table 3 and are compared to the ones of complex **14**. While [CuCl(H-CDP(CH_2_PPh_2_)_2_]PF_6_ (**17**) can be considered as a complex of a cationic ligand, [CuPPh_3_(CH(PPh_2_CHPPh_2_)_2_] (**18**) has to be considered as an example of a complex with the deprotonated, anionic form of ligand **13**. The charge distribution of the corresponding ligand is also reflected in the C–P distances within the complexes **14**, **17**, and **18**. While C1–P1 and C1–P2 are distinctly shorter for **14**, an increase in C–P bond distance is observed for **17** and **18** due to the protonation of C1. Furthermore, the deprotonation of C2 and C3 of complex **18** leads to a shortening of the distances C2–P1, C2–P3 and C3–P2, C3–P4 compared to **14** and **17**, where C2 and C3 are considered as methylene groups. This also corresponds to the P1–C2–P3 and P2–C3–P4 angles, which are significantly larger for the anionic ligand complex **18** compared to **14** and **17**. 

### 2.3. Photophysical Characterization of Selected CDP Complexes

Since the photophysical properties of carbodiphosphorane Cu(I) complexes have not yet been considered, the first investigations were performed in this report. The Cu(I) complexes **2**–**7**, **9** and **14**–**16** show photoluminescence upon irradiation with UV light at room temperature. As proof of concept, we investigated emission spectra and quantum yields of [(CuPPh_3_)_2_(CDP(Py)_2_)](PF_6_)_2_ (**4**) and [(CuSPh)_2_(CDP(CH_2_PPh_2_)_2_] (**16**). Figure 3 illustrates the normalized room-temperature emission spectra of these materials. Compound **4** shows an emission maximum at 541 nm, corresponding to green/yellow color, along with a quantum yield (*Φ*_PL_) of 36% for the powder sample. The emission maximum of [(CuSPh)_2_(CDP(CH_2_PPh_2_)_2_] (**16**) (powder) is found at 510 nm (green color) showing *Φ*_PL_ = 60%. The high quantum yields indicate the relatively high rigidity of the complexes in powder form. Moreover, these materials are chemically robust: After exposing the complexes to air for two months, the compounds still show their characteristic photoluminescence upon irradiation with UV light at room temperature. 

First insight in the electronic structure of the emitting compounds **4** and **16** is obtained from consideration of the HOMO and LUMO distributions. Figure 4 displays that the HOMO shows for both compounds significant participation of metal d character as well as a marginal contribution of the central carbon. The LUMO, on the other hand, is primarily localized at the pyridyl units of the ligand backbone as well as on the phenyl groups attached to P1 and P2 for **16**. Considering HOMO→LUMO transitions, the excitations can be ascribed to metal-to-ligand charge transfer (MLCT) transitions. According to the relatively small HOMO-LUMO overlap, it is indicated that the energy separations *ΔE*(S_1_–T_1_) between the lowest singlet S_1_ and triplet T_1_ states are small enough to allow for up-inter-system crossing at ambient temperature [28,31,32]. Therefore, we tentatively assign the emission observed as TADF emission. Details will be reported in a subsequent study. 

For completeness, it is mentioned that also complexes **17** and **18** exhibit photoluminescence upon irradiation with UV light at room temperature. This was not the case for [Cu_2_(dppf)(CDP(Py)_2_)](PF_6_)_2_ (**8**) and [(CuCarb)_2_(CDP(Py)_2_)] (**11**). For **8**, quenching of the ferrocenyl ligand could be responsible for the lack of photoluminescence. In case of **11**, a reason could be the asymmetric coordination found in the crystal structure. The reduced rigidity could lead to larger geometry rearrangement after excitation and thus to quenching.

## 3. Conclusions

We successfully isolated and characterized a series of dinuclear copper(I) complexes of two so far poorly investigated, multidentate pyridyl and phosphanyl functionalized *N,C,N-* and *P,C,P-* carbodiphosphorane ligands. A series of neutral complexes of CDP(Py)_2_ (**1**) with anionic coligands X and a series of dicationic complexes with monodentate and bridging bidentate bisphosphine ligands DPEPhos, XantPhos, and dppf were fully characterized, including their XRD molecular structures. In order to prepare unprecedented dinuclear copper complexes with a previously discovered *P,C,P*-carbodiphosphorane ligand backbone, it was necessary to isolate the free ligand base CDP(CH_2_PPh_2_)_2_ (**13**), which has not been demonstrated before. **13** can be obtained from [CH(dppm)_2_]Cl (**12**) and an excess of sodium amide in liquid ammonia. DFT calculations reveal that the ground state of **13** has no CDP structure in the gas phase, but rather an unsymmetric tautomer form **13a**. However, upon reaction with CuX, the CDP tautomer is trapped from the tautomeric equilibrium and neutral dinuclear Cu(I) CDP complexes are isolated and fully characterized. In addition, a protonated and a deprotonated ligand form of **13** was characterized in mononuclear complexes [CuCl(H-CDP(CH_2_PPh_2_)_2_]PF_6_ (**17**) and [CuPPh_3_(CH(PPh_2_CHPPh_2_)_2_] (**18**). With the exception of [Cu_2_(dppf)(CDP(Py)_2_)](PF_6_)_2_ (**8**) and [(CuCarb)_2_(CDP(Py)_2_)] (**11**)), the complexes studied show photoluminescence upon irradiation with UV light at room temperature. Photophysical measurements reveal quantum yields *Φ*_PL_ of 36% and 60% for [(CuPPh_3_)_2_(CDP(Py)_2_)](PF_6_)_2_ (**4**) and [(CuSPh)_2_(CDP(CH_2_PPh_2_)_2_] (**16**). As found in the crystal structure, the formal central carbon(0) atom is capable of coordinating two copper atoms relatively close to each other. They are further coordinated in a chelating manner to the chelating side arms of the CDPs. This rigid ligand design leads to high-emission quantum yields and makes the CDP complexes relatively stable under air. Therefore, it is proposed to test the compound’s OLED suitability.

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
