# Peer review of "Cu(I) Complexes of Multidentate N,C,N- and P,C,P-Carbodiphosphorane Ligands and Their Photoluminescence"

_molecules, 2020, doi:10.3390/molecules25173990_

Round 1
Reviewer 1 Report
Th emanuscript submitted by Sundermeyer et al introduces the carbodiphosphorane ligand to Cu(I) chemistry and reports a photoluminescence study.
The manuscript is well ordered and well written and the topic of the science is interesting, albeit nothing earth shattering, the authors do a good job and it was a joy to read. I only have a few minor comments, detailed below and I recommend that this manuscript be accepted after these revisions.
- You measured the 13C{1H} NMR spectra so please correct throughout the manuscript and SI.
- In the experimental part of the SI, you have written "13C NMR (CCl2D,..." you are missing the subsript 2.
- For the PL measurements there is some information missing. For example what does the excitation spectrum look like, especially compared to the UV-Vis spectra? What are the lifetimes, if these can be measured?
Author Response
- You measured the 13C{1H} NMR spectra so please correct throughout the manuscript and SI.
A: 13C NMR was changed to 13C{1H} NMR
- In the experimental part of the SI, you have written "13C NMR (CCl2D,..." you are missing the subsript 2.
A: CCl2D has been changed to CCl2D2. - For the PL measurements there is some information missing. For example what does the excitation spectrum look like, especially compared to the UV-Vis spectra? What are the lifetimes, if these can be measured?
A: Currently, more experimental data are not available. More photophysical studies will be following in planned investigations including OLED device fabrications in the future.
Reviewer 2 Report
In this manuscript the authors report the preparation, characterization and luminescence properties of a series of Cu(I) complexes of N,C,N- and P,C,P-2-carbodiphosphorane ligands. These ligands are of interest for their bonding interactions with metals and ability to produce rigid luminescent compounds.
This is a synthesis and characterization manuscript. Overall, the work is well done and I am supportive of publication in Molecules. However, minor corrections to improve the manuscript are the following:
1) The use of the abbreviation [M] in Scheme 1 (structural drawings of 2 and 3) and throughout the manuscript is confusing as other axial ligands (e.g., -SC6F5) are shown.
2) The bond distances in various structures in the manuscript and SI are inconsistently drawn. For example, see drawing of 17 in Scheme 2 - very short Cu-Cl bond length.
3) In the caption for Figure 1 is should be noted that only the cationic portions of 4, 6 and 7 are shown. Same for the caption for 17 in Figure 2.
4) The dynamic exchange process of the pyridyl ligands in 11 should be investigated using variable temperature 1H NMR. Can an exchange rate be determined using 1H NMR? No spectral evidence of this exchange interaction is provided in the SI.
5) At various places in the manuscript, "where" is used instead of "were". For example, p. 5, line 170. Also, p. 6, line 217. Others are present as well.
6) p. 5, line 174: A Lewis acid activated acetonitrile ligand?
7) p. 7, line 220: Chapter 2.1?
8) p. 8, line 251: "considered" not "considers"
9) Figure 3: Wavelength is spelled wrong on the X-axis.
10) Global problems in SI:
a) Elemental analysis C,H,N data outside 0.4% from predicted value with no explanation. Are these materials analytically pure?
b) No elemental analysis data is provided for 14-17. Are these materials analytically pure?
b) Significant figures are not well done.
c) Structures should be drawn on 1H and 13C NMR spectra showing assignments, if possible. Some spectra are assigned on the spectra, others are not.
d) 13C data is sometimes shown without a listing of the resonances in the experimental characterization data. Listing of a number of resonances expected and observed in the 13C spectra would also be useful.
e) Why is an IR spectrum provided only for 4?
Author Response
In this manuscript the authors report the preparation, characterization and luminescence properties of a series of Cu(I) complexes of N,C,N- and P,C,P-2-carbodiphosphorane ligands. These ligands are of interest for their bonding interactions with metals and ability to produce rigid luminescent compounds.
This is a synthesis and characterization manuscript. Overall, the work is well done and I am supportive of publication in Molecules. However, minor corrections to improve the manuscript are the following:
1) The use of the abbreviation [M] in Scheme 1 (structural drawings of 2 and 3) and throughout the manuscript is confusing as other axial ligands (e.g., -SC6F5) are shown.
A: Scheme 1 has been reconsidered.
2) The bond distances in various structures in the manuscript and SI are inconsistently drawn. For example, see drawing of 17 in Scheme 2 - very short Cu-Cl bond length.
A: Scheme 2 has been reconsidered.
3) In the caption for Figure 1 is should be noted that only the cationic portions of 4, 6 and 7 are shown. Same for the caption for 17 in Figure 2.
A: The description of the missing counter anion has been added to the caption.
4) The dynamic exchange process of the pyridyl ligands in 11 should be investigated using variable temperature 1H NMR. Can an exchange rate be determined using 1H NMR? No spectral evidence of this exchange interaction is provided in the SI.
A: Unfortunately, no exchange rate can be determined. Temperature dependent NMR spectroscopy did not give much evidence of this exchange interaction. But no lower temperature could be measured due to the solubility of the compound 11. A figure displaying the temperature dependence of proton nmr spectra of 11 has been added to the electronic supplement as Fig. S-25.
5) At various places in the manuscript, "where" is used instead of "were". For example, p. 5, line 170. Also, p. 6, line 217. Others are present as well.
A: Where has been changed to were.
6) p. 5, line 174: A Lewis acid activated acetonitrile ligand?
A: lewis has been changed to Lewis.
7) p. 7, line 220: Chapter 2.1?
A: chapter has been changed to Chapter.
8) p. 8, line 251: "considered" not "considers"
A: "considers" has been changed to considered".
9) Figure 3: Wavelength is spelled wrong on the X-axis.
A: Figure 3 has been adjusted.
10) Global problems in SI:
a) Elemental analysis C,H,N data outside 0.4% from predicted value with no explanation. Are these materials analytically pure?
A: As a matter of fact, the combustion analysis of air sensitive metal containing compounds is sometimes tricky in our departmental analytic lab: We learnt from experimental deviations observed in the range 0.5 up to 1,5% from one and the same phase pure, single crystalline metal organic sample, that such deviations - rather untypical for purely organic non-air sensitive reference compounds - have to be expected in some cases. Deviations are dependent on the time between weighting the 4 mg sample into the vial (under a blanket of argon), putting the vial into the vial holder of the sample carousel and the delay time of its combustion. Finally, they are dependent on the type of metal, the combustion temperature, additives for better combustion and for less formation of metal carbides and nitrides. As we have isolated nearly all compounds in single crystalline phase and have characterized even more than those reported here via XRD analysis, we are absolutely confident, that all compounds described herein were isolated in analytically phase pure form – sometimes with additional solvent molecules in the lattice, difficult to eliminate under vacuum and higher temperatures.
We added this comment to the supplement.
b) No elemental analysis data is provided for 14-17. Are these materials analytically pure?
A: Off course: large crops of phase pure single crystals of 14-17 are analytically pure. The costly combustion analysis was done not routinely for all the compound – especially after the XRD analysis had already been conducted.
c) Significant figures are not well done.
A: we fixed what we could identify un this respect.
d) Structures should be drawn on 1H and 13C NMR spectra showing assignments, if possible. Some spectra are assigned on the spectra, others are not.
A: Assignment was only considered if the spectrum was clearly assignable.
e) 13C data is sometimes shown without a listing of the resonances in the experimental characterization data. Listing of a number of resonances expected and observed in the 13C spectra would also be useful.
A: A: The essential resonances are shown in the experimental characterization data.
f) Why is an IR spectrum provided only for 4?
A: IR has only been measured exemplarily for compound 4. There is no reason. IR is not essential for the characterization of this class of compounds.
Reviewer 3 Report
Sundermezer and co-workers have delivered a thorough and complete study on the coordination chemistry of two particular CDP ligands with Cu(I) precursors, including some preliminary photophysical data to underline the potential of these systems for photohemical applications.
The main body of the work revolves around spectroscopic and X-ray crystallographic characterization of the various complexes, including a nice investigation of base dependent deprotonation of bis(dppm)C, which is also supported by DFT studies.
This paper is ready for publication after the following minor points are addressed.
minor typo: Lewis acid, not lewis acid (page 5, line 174)
the clarity of various XRD pictures could be improved. The note that labelling is identical for all species appears missing.
some of the elemental analysis appears off in C - please elaborate
Introduction could do with a graphic to depict the state of art with CDPs
Complex 18 (SI) is reported to show 3 doublets in the 31P NMR but two different J's are reported... where is the counterpart for the 118 Hz coupling constant? General: 31P not assigned? Integration?
Figures S1, S16 and S32 show impure 31P NMR specta
Author Response
Sundermeyer and co-workers have delivered a thorough and complete study on the coordination chemistry of two particular CDP ligands with Cu(I) precursors, including some preliminary photophysical data to underline the potential of these systems for photohemical applications.
The main body of the work revolves around spectroscopic and X-ray crystallographic characterization of the various complexes, including a nice investigation of base dependent deprotonation of bis(dppm)C, which is also supported by DFT studies.
This paper is ready for publication after the following minor points are addressed.
- minor typo: Lewis acid, not lewis acid (page 5, line 174)
A: lewis has been changed to Lewis. - the clarity of various XRD pictures could be improved. The note that labelling is identical for all species appears missing.
A: The XRD structures are very complex and the molecules consists of many atoms with a tendency of functionalities overlapping in the 2D projection. We tested many ways to depict the molecular structures. The pictures shown were found be to best suitable for publication. The note that labelling is identical for all species has been added to the caption. - some of the elemental analysis appears off in C - please elaborate
A: As a matter of fact, the combustion analysis of air sensitive metal containing compounds is sometimes tricky in our departmental analytic lab: We learnt from experimental deviations observed in the range 0.5 up to 1.5% from one and the same phase pure, single crystalline metal organic sample, that such deviations - rather untypical for purely organic non-air sensitive reference compounds - have to be expected in some cases. Deviations are dependent on the time between weighting the 4 mg sample into the vial (under a blanket of argon), putting the vial into the vial holder of the sample carousel and the delay time of its combustion. Finally, they are dependent on the type of metal, the combustion temperature, additives for better combustion and for less formation of metal carbides and nitrides. As we have isolated nearly all compounds in single crystalline phase and have characterized even more than those reported here via XRD analysis, we are absolutely confident, that all compounds described herein were isolated in analytically phase pure form – sometimes with additional solvent molecules in the lattice, difficult to eliminate under vacuum and higher temperatures. We added this comment in the supplement. - Introduction could do with a graphic to depict the state of art with CDPs
A: As there are two reviews on CDPs published recently, we decided not to expand this 14 pages long article with its 74 references and more than 50 pages of supplement into a third review. - Complex 18 (SI) is reported to show 3 doublets in the 31P NMR but two different J's are reported... where is the counterpart for the 118 Hz coupling constant? General: 31P not assigned? Integration?
A: This coupling arises from the interactions of the 31P nuclei with the 63,65Cu isotopes. - Figures S1, S16 and S32 show impure 31P NMR spectra
A: Protonation of the CDPs is sometimes a characteristic side reaction;
therefore traces of the protonated species are sometimes observed, even
after recrystallisation.
Reviewer 4 Report
The authors synthesized Cu complex with multidentate carbodiphosphrane ligands. The discussion of the structure of the intermediate was also involved. And they investigated the fluorescence properties of those complexes. But I could not accept for the result of some analysis in this manuscript. Therefore, I cannot recommend this manuscript to publish in Molecules at this stage. If the authors wish to revise, please consider the following comments.
1) In lines 103-107, the authors assigned the m/z (539.18016) as the ion peak of compound 11. However, the exact mass of compound 11 (C59H44N4P2Cu2) should be 996.1628 ([M]+). I think that the number of m/z 539.18016 would be compound 1 (C35H29N2P2, [M+H]+ 539.1806).
2) From line 186, the authors discussed the structure of compound 13 based on the DFT calculation and 31P NMR spectrum. The authors commented the 31P NMR of compound 13 was “highly complex”. But the authors also assigned as “31P{H} NMR (C6D6, 101 MHz): 23.2 (dt, J = 8.0, 14.7 Hz), -15.8 (d, J = 118.5 Hz)” in supporting information. Those comments seem to contradict, I think. I also wonder why the coupling with triplet is occurred. Detailed investigation of that spectrum, I can also find the three peaks around -15.8 ppm as two shoulders. According to those findings, there is three types of structures in this spectrum. Furthermore, if the ratio of 13a, 13b, and 13c calculated from differential of energy were accordance with the ratio from the spectrum, the authors asserting would be acceptable. And if those three peaks were attributed to the structural isomers, the ratio would be changed in the different temperatures. When the peak shape could not be changed in the different temperature, there is single compound or non-equilibrium mixtures.
3) From line 285, the authors tentatively assigned that the emission of compound 4 and 16 was based on TADF. But no information to support that postulation was shown. In the manuscript, the authors commented the small ΔE(S1 – T1). But the data was not found. I strongly recommend that some facts to clarify the nature of that emission should be shown, or to withdraw the comment about TADF.
4) In line 92, the compound number should be 4-8 not 4-9. Please revise it.
5) In Scheme 2, reagents toward compound 17 and compound 18 ([Cu(NCMe)4]PF6) should be written properly. And cationic plus character should be shown in compound 17.
6) In Figures S-29, S-30, S-32, the figures of original spectrum and inset spectrum were different. Please revise them.
Author Response
The authors synthesized Cu complex with multidentate carbodiphosphorane ligands. The discussion of the structure of the intermediate was also involved. And they investigated the fluorescence properties of those complexes. But I could not accept for the result of some analysis in this manuscript. Therefore, I cannot recommend this manuscript to publish in Molecules at this stage. If the authors wish to revise, please consider the following comments.
1) In lines 103-107, the authors assigned the m/z (539.18016) as the ion peak of compound 11. However, the exact mass of compound 11 (C59H44N4P2Cu2) should be 996.1628 ([M]+). I think that the number of m/z 539.18016 would be compound 1 (C35H29N2P2, [M+H]+ 539.1806).
A: Thank you for your care: This point has been carefully considered and adjusted to the manuscript.
2) From line 186, the authors discussed the structure of compound 13 based on the DFT calculation and 31P NMR spectrum. The authors commented the 31P NMR of compound 13 was “highly complex”. But the authors also assigned as “31P{H} NMR (C6D6, 101 MHz): 23.2 (dt, J = 8.0, 14.7 Hz), -15.8 (d, J = 118.5 Hz)” in supporting information. Those comments seem to contradict, I think. I also wonder why the coupling with triplet is occurred. Detailed investigation of that spectrum, I can also find the three peaks around -15.8 ppm as two shoulders. According to those findings, there is three types of structures in this spectrum. Furthermore, if the ratio of 13a, 13b, and 13c calculated from differential of energy were accordance with the ratio from the spectrum, the authors asserting would be acceptable. And if those three peaks were attributed to the structural isomers, the ratio would be changed in the different temperatures. When the peak shape could not be changed in the different temperature, there is single compound or non-equilibrium mixtures.
A: We changed our wording: After deprotonation of symmetric protonated CDP form 12, 31P{1H} NMR spectra of the product (or products) become temperature and solvent dependent. We presumed, that this observation could be an indication of the presence of more than one tautomer, at least two with definitely chemically non-equivalent 31P nuclei of mono-deprotonated base 13 (see Figure S-27). As there were no literature data available on this particular carbodiphosphorane 13, despite of the fact, it was used as ligand in several publications, we decided to investigate the tautomeric forms of 13 via computational methods (Scheme 3).
…... Therefore, 13b seems to be observable at very low concentration in dynamic equilibrium ratio next to 13a but not symmetric carbodiphosphorane form 13c.
3) From line 285, the authors tentatively assigned that the emission of compound 4 and 16 was based on TADF. But no information to support that postulation was shown. In the manuscript, the authors commented the small ΔE(S1 – T1). But the data was not found. I strongly recommend that some facts to clarify the nature of that emission should be shown, or to withdraw the comment about TADF.
A: According to the extensive investigations available for Cu(I) TADF compounds, it may well be concluded that very small orbital overlap of HOMO and LUMO is connected with a small ΔE(S1 – T1) gap. [28, 31, 32] Accordingly, efficient RISC will usually occur. Hence, the ambient temperature emission may tentatively be assigned as TADF. Unfortunately, other spectroscopic data are currently not available.
4) In line 92, the compound number should be 4-8 not 4-9. Please revise it.
A: 4-9 has been changed to 4-8.
5) In Scheme 2, reagents toward compound 17 and compound 18 ([Cu(NCMe)4]PF6) should be written properly. And cationic plus character should be shown in compound 17.
A: Scheme 2 has been adjusted.
6) In Figures S-29, S-30, S-32, the figures of original spectrum and inset spectrum were different. Please revise them.
A: The spectra have been adjusted.
Round 2
Reviewer 4 Report
I can accept this manuscript to publish in Molecules at this version.